# Pilot Study of Anti-Th2 Immunotherapy for the Treatment of Breast Cancer-Related Upper Extremity Lymphedema

**DOI:** 10.3390/biology10090934

**Published:** 2021-09-18

**Authors:** Babak J. Mehrara, Hyeung Ju Park, Raghu P. Kataru, Jacqueline Bromberg, Michelle Coriddi, Jung Eun Baik, Jinyeon Shin, Claire Li, Michele R. Cavalli, Elizabeth M. Encarnacion, Meghan Lee, Kimberly J. Van Zee, Elyn Riedel, Joseph H. Dayan

**Affiliations:** 1Plastic and Reconstructive Surgery Service, Department of Surgery, Memorial Sloan Kettering Cancer Center, New York, NY 10065, USA; parkh@mskcc.org (H.J.P.); Katarur@mskcc.org (R.P.K.); Coriddim@mskcc.org (M.C.); baikj1@mskcc.org (J.E.B.); shinj1@mskcc.org (J.S.); yul9025@nyp.org (C.L.); mcavalli127@gmail.com (M.R.C.); encarnae@mskcc.org (E.M.E.); meghan.e.lee@gmail.com (M.L.); dayanj@mskcc.org (J.H.D.); 2Breast Medicine Service, Department of Medicine, Memorial Sloan Kettering Cancer Center, New York, NY 10065, USA; bromberj@MSKCC.ORG; 3Breast Surgery Service, Department of Surgery, Memorial Sloan Kettering Cancer Center, New York, NY 10065, USA; vanzeek@mskcc.org; 4Department of Epidemiology & Biostatistics, Memorial Sloan Kettering Cancer Center, New York, NY 10065, USA; riedele@mskcc.org

**Keywords:** lymphedema, immunotherapy, skin, Th2 inflammation, breast cancer, keratinocytes

## Abstract

**Simple Summary:**

Lymphedema is a common complication of cancer, and patients with lymphedema have substantially decreased quality of life and suffer from lifelong symptoms. This study aimed to examine the effect of inhibition of Th2 inflammation in lymphedema by using QBX258, a combination of IL4/13 neutralizing antibodies. QBX258 treatment increased quality of life and reduced pathologic changes in skin including hyperkeratosis, cytokine production, fibrosis and immune cell recruitment. In conclusion, this study suggested that immunotherapy against IL4/13 improved patients’ daily life which might be related with reduced pathological skin changes.

**Abstract:**

Recent studies suggest that Th2 cells play a key role in the pathology of secondary lymphedema by elaborating cytokines such as IL4 and IL13. The aim of this study was to test the efficacy of QBX258, a monoclonal IL4/IL13 neutralizing antibody, in women with breast cancer–related lymphedema (BCRL). We enrolled nine women with unilateral stage I/II BCRL and treated them once monthly with intravenous infusions of QBX258 for 4 months. We measured limb volumes, bioimpedance, and skin tonometry, and analyzed the quality of life (QOL) using a validated lymphedema questionnaire (Upper Limb Lymphedema 27, ULL-27) before treatment, immediately after treatment, and 4 months following treatment withdrawal. We also obtained 5 mm skin biopsies from the normal and lymphedematous limbs before and after treatment. Treatment was well-tolerated; however, one patient with a history of cellulitis developed cellulitis during the trial and was excluded from further analysis. We found no differences in limb volumes or bioimpedance measurements after drug treatment. However, QBX258 treatment improved skin stiffness (*p* < 0.001) and improved QOL measurements (Physical *p* < 0.05, Social *p* = 0.01). These improvements returned to baseline after treatment withdrawal. Histologically, treatment decreased epidermal thickness, the number of proliferating keratinocytes, type III collagen deposition, infiltration of mast cells, and the expression of Th2-inducing cytokines in the lymphedematous skin. Our limited study suggests that immunotherapy against Th2 cytokines may improve skin changes and QOL of women with BCRL. This treatment appears to be less effective for decreasing limb volumes; however, additional studies are needed.

## 1. Introduction

Secondary lymphedema is the most common long-term complication of cancer treatment, particularly following axillary lymph node dissection for breast cancer (15–50% of patients) and treatment of other solid tumors (15–20% of patients), such as melanoma, sarcoma, and gynecological and urologic malignancies [1,2,3,4]. The number of patients with this lifelong disease increases each year due to improved long-term survival following cancer treatment and rising rates of important risk factors for the disease such as obesity, adjuvant radiation therapy, and advanced age [5,6,7]. 

Although lymphedema is a highly morbid, progressive disease that significantly impairs function and quality of life, there is no medical or surgical cure. Instead, patients are treated with palliative measures using compression garments and physical therapy to prevent disease progression and relieve symptoms [8,9,10,11]. These treatments are time-consuming and expensive, leading to a high degree of non-compliance and disease progression [9]. Surgical treatments for lymphedema have recently been developed and are helpful in some patients, but these treatments are invasive and not very effective for patients with advanced lymphedema. 

The development of novel therapies for secondary lymphedema has been limited because the pathophysiology of this disease is poorly understood. We know that lymphatic injury resulting from cancer surgery is a critical initiator of this disease, but it is not clear why some patients develop lymphedema while others do not. Although many studies have identified clinical risk factors for lymphedema, the mechanisms by which these comorbid conditions modulate the pathophysiology of the disease are less well-defined. Perhaps most intriguingly, it is not fully understood why, in most cases, lymphedema develops in a delayed manner, months or years after surgery [12]. These epidemiologic features suggest that additional pathologic events (i.e., “second hits”) are necessary for lymphedema to develop. Therefore, identifying the sequence of events that translates lymphatic injury to lymphedema may provide effective therapeutic targets. 

Several research groups have hypothesized that chronic inflammation may be an important regulator in the pathological sequence of lymphedema development. Although chronic inflammation is a histologic feature of lymphedema, Rockson and colleagues were the first to demonstrate that persistent inflammation may also serve as a critical pathologic driver of the disease [13]. Our group also analyzed clinical lymphedema biopsy specimens and mouse models of lymphedema to identify the inflammatory signature of lymphedema. By comparing the normal and lymphedematous limbs of women with unilateral breast cancer–related lymphedema (BCRL), we found that the severity of lymphedema, increasing International Society of Lymphology stage, positively correlates with the number of CD4^+^ Th cells in the dermis and subcutaneous tissues of the lymphedematous limb [14]. In other studies, we showed that transgenic mice lacking Th cells, or inhibition of Th cell responses using neutralizing antibodies or tacrolimus, a calcineurin inhibitor that decreases T cell proliferation by antagonizing IL2, can both prevent the development of lymphedema and treat it once it has developed [15,16,17]. In contrast, depletion of cytotoxic T cells, macrophages, or B-cells had no effect or worsened the phenotype of lymphedema [15,16,18]. More recently, other research groups have also shown that Th cells play a crucial role in the pathophysiology of lymphedema, confirming our previous findings [19,20]. 

We also found that Th2 differentiation is an important regulator of lymphedema development [14,21]. Clinical biopsy specimens from patients with unilateral BCRL showed an increased number of Th2 cells infiltrating into the dermis and subcutaneous tissues of lymphedematous tissues compared with normal skin [14]. Using mouse models, we showed that shortly following lymphatic injury, DCs and Langerhans cells just below the epidermis are activated and migrate to the regional lymph nodes [16]. There, these antigen-presenting cells promote differentiation of naïve Th cells to a mixed Th1/Th2 phenotype that expresses skin-homing receptors that guide them to the skin distal to the zone of lymphatic injury. Transgenic mice that have impaired Th2 differentiation capability do not develop lymphedema following lymphatic injury [21]. In contrast, mice with impaired Th1 differentiation capacity develop lymphedema, and the severity of the disease is indistinguishable from wild-type controls. Similarly, treatment of mice with neutralizing antibodies against IL4 or IL13, cytokines necessary for differentiation of naïve Th cells to Th2 phenotype, not only prevented the development of lymphedema in preclinical mouse models but also treated lymphedema once it was established [14]. Mechanistically, we found that cytokines derived from Th2 cells (IL4, IL13) increase lymphatic leakiness, increase collagen deposition and fibrosis in the dermis and collecting lymphatics, inhibit lymphangiogenesis, and impair lymphatic pumping [14,22]. These findings led us and others to hypothesize that lymphedema is a fibroproliferative disease with the progressive replacement of functional parenchyma (lymphatic vessels) with fibrous tissues [20]. This hypothesis is supported by the fact that fibroproliferative conditions are common; fibrosis is a preserved end-pathway for chronic inflammation and organ failure in the heart, lung, pancreas, skin, and kidney [23]. The fibroproliferative hypothesis also explains why lymphedema develops in a delayed fashion following surgery since the fibrotic threshold needed to develop symptoms takes time to happen. Variability in fibrotic responses among patients may also provide a rationale for the clinical observation that the severity of secondary lymphedema is highly variable and difficult to predict. Thus, inhibition of fibrosis with anti-Th2 therapies may be a viable means of clinically preventing or treating lymphedema. 

Targeted therapeutics that inhibit immune responses are now commonly used to treat a variety of chronic disorders, such as atopic dermatitis, psoriasis, ankylosing spondylitis, inflammatory bowel disease, asthma, and rheumatoid arthritis. These treatments have significantly reduced treatment costs and improved outcomes by decreasing side effects of traditional anti-inflammatory therapies such as corticosteroids and non-specific immunosuppressants. In most cases, monoclonal antibodies inhibit cytokines or cytokine receptors, thereby blocking the inflammatory pathway responsible for the disease. This approach has also been used for Th2 disorders by blocking the activity of both IL4 and IL13 since these cytokines share a common receptor (IL4-Rα) and have overlapping biologic activity. Dupilumab, a monoclonal antibody that binds the alpha chain of IL4-Rα, is FDA approved for the treatment of adult patients with moderate-severe atopic dermatitis who have failed treatment with topical agents [24]. This drug has also shown promising results with chronic sinusitis and nasal polyposis [25] and bullous pemphigoid [26]. Other uses of IL4 and IL13 antibodies have been developed and used in clinical trials for Th2-mediated diseases, including eosinophilic esophagitis, some forms of asthma, and pulmonary fibrosis. Based on this rationale, the purpose of this study was to assess the safety and efficacy of a combination treatment with monoclonal antibodies that block both IL4 and IL13 (QBX258) for the treatment of patients with BCRL.

## 2. Materials and Methods

### 2.1. Study Design and Approval

This pilot study was approved by the IRB at the Memorial Sloan Kettering Cancer Center. Written informed consent was obtained from all eligible participants prior to inclusion in the study. Our study was a single-arm, open-label pilot study designed to test the safety of QBX258, a combination of two humanized monoclonal antibodies that inhibit the bioactivity of IL4 (VAK296) and IL13 (QAX576), for the treatment of stage I or II BCRL. Women ages 18 to 70 who were diagnosed with unilateral stage I (spontaneously reversible) or II (spontaneously irreversible) BCRL, as defined by the International Society of Lymphology [27], were screened for inclusion in the study. The inclusion and exclusion criteria for the study are listed in Table 1. Patients who qualified for the study underwent measurement of arm volume differences. Those with a minimum volume excess of 300 mL compared with the normal upper extremity were offered the opportunity to join the trial. The study design is summarized in Table 2.

We elected to limit the study to patients with stage I or II lymphedema since previous studies have shown that antifibrotic strategies are most effective in early-stage disease (i.e., when end-organ failure is not complete) [23]. We chose not to include patients with latent (stage 0) lymphedema since these patients typically have very little difference in arm volumes between the lymphedematous and normal limbs, thus making it challenging to analyze the relative effectiveness of our treatment on our primary clinical outcome measure (volume reduction). Similarly, we chose not to include patients with end-stage (stage III) lymphedema since it is unlikely that the pathologic changes that have occurred in these cases (usually for many years) will be reversed with the short-term treatment proposed in this study.

### 2.2. Pretreatment Evaluation and Measurements 

Arm volume measurements were performed in duplicate at baseline using a perometer (Pero-Systems; Wuppertal, Germany). A licensed physical therapist performed measurements between the hours of 8 a.m. and 3 p.m. The perometer is a non-invasive infrared scanner that measures the circumference of the upper extremity at multiple points; volumes are then calculated using the truncated cone formula. This analysis is highly reproducible and accurate, avoiding the need for traditional volume measurements (fluid displacement) or manual limb circumference measurements. The perometer has excellent intra-rater/inter-rater and test-retest reliability (intraclass correlation coefficient = 0.997) with a greater than 95% specificity for lymphedema [28,29]. The volume excess was calculated and compared with the normal (i.e., contralateral) upper extremity. 

Bioimpedance measurements were performed in duplicate to compare the normal and lymphedematous limbs using the ImpediMed L-Dex U400 device (Carlsbad, CA, USA) [30,31]. Bioimpedance has very high sensitivity (100%) and specificity (98%) for early-stage lymphedema [32].

Skin fibrosis was assessed in duplicate at a point located 10 cm above and 10 cm below the olecranon process on the dorsal and ventral surfaces of the lymphedematous and normal limb using a tonometer (ElastiMeter; Delfin Technologies, FL, USA). Skin tonometry measurements are a sensitive means of analyzing the response to treatment and disease progression in patients with lymphedema [33,34]. The ICC for test-retest and interrater reliability of tonometry ranges from 0.69 to 0.88, and tonometry is considered to have very high reliability [35]. The ElastiMeter is a battery-operated device that measures the force required to indent the skin at a given location. The indentation that is measured is minor and monitored with built-in force sensors that measure skin elasticity based on a low-level deformation force that enables instantaneous measurements. 

A quality of life survey (Upper Limb Lymphedema 27, ULL-27) was administered upon enrollment and at various times during the study to quantify the subjective measures of lymphedema. The ULL-27, validated for patients with lymphedema, is simple to complete and enabled us to track patient-reported outcomes of lymphedema [36,37]. The ULL-27 is composed of 27 items divided into 3 dimensions: ‘physical’ (15 items), ‘psychological’ (7 items), and ‘social.’ (5 items). The recall period is short (previous 4 weeks), enabling us to compare pre- and post-treatment quality of life measures. The ULL-27 has a high degree of sensitivity for quantifying clinical improvements in lymphedema symptoms [38]. Lower scores on the ULL-27 indicate increasing degrees of impairment. 

### 2.3. Treatment Plan

Once registered, patients were treated with QBX258 (VAK296 [3 mg/kg] and QAX576 [6 mg/kg]) delivered via peripheral intravenous infusion once every 4 weeks (±1 week) for 4 treatments. Intravenous access was limited to the non-lymphedematous limb and was obtained using standard techniques. Patients were monitored for 2 h after infusion to document infusion reactions or other adverse events. Repeat measurements were performed within 21 days of the final treatment dose. Washout measurements were performed 16–20 weeks after the last dose of QBX258 to analyze changes after treatment withdrawal. Because the treatment half-life of QBX258 is 4 weeks, a washout period of 16–20 weeks after the last treatment is sufficient for complete clearance of the drug. 

During the treatment and washout periods, patients continued their usual (i.e., pre-enrollment) routine for lymphedema care, including the use of skincare products and compression garments. However, to avoid potential confounding effects, patients avoided the use of intermittent pneumatic compression devices, manual lymphatic massage by a physical therapist, laser treatment, or treatment with high compression short stretch bandages during the study period. 

### 2.4. Outcomes 

Our primary objectives were to assess the safety of QBX258 and preliminarily determine the efficacy of this treatment on reducing arm volume excess. Our secondary objectives were to analyze the effectiveness of QBX258 in decreasing fluid content of the affected extremity (bioimpedance measurements), examine changes in skin elasticity/fibrosis in response to treatment (skin tonometry), analyze changes in quality of life measures (ULL-27), and to quantify histologic changes in lymphedematous tissues (skin biopsies before and after treatment). Outcome measures were assessed within 21 days of the last dose of QBX258 and after a washout period of 16–20 weeks.

#### 2.4.1. Calculation of Arm Volume Changes

The difference in the extent of lymphedema before and after treatment was assessed with paired 2-sided *t*-tests comparing the post-treatment and washout measurements with the pretreatment measurements and the post-treatment measurements with the washout measurements. Therapeutic volume changes in the arm were calculated using the methods published by Anderson et al. [39]. Briefly, the difference in volume measurements between the normal and lymphedematous arms at baseline (i.e., volume excess) was compared with the volume differential after drug treatment and following the washout period using the following formula: (V_L_ − V_N_) _B_ − (V_L_ − V_N_) _F_
V_L_ = Volume of the lymphedematous arm
V_N_ = Volume of the normal arm
B = Baseline
F = Follow up

#### 2.4.2. Analysis of Secondary Objectives

ULL-27 questionnaire responses were graded and scored using previously published methods. Changes in each symptom scale (physical, psychological, and social) (Table 3) and the overall score were compared pre- and post-treatment using paired 2-sided *t*-tests [38]. The bioimpedance and tonometry measurements were continuous variables and were analyzed using paired *t*-tests. 

#### 2.4.3. Histologic Analysis

Full-thickness 5-mm skin punch biopsies were harvested under sterile conditions from the volar surface of the affected and normal arms at a point 5–10 cm above or below the elbow. The exact location of skin biopsy was based on a physical exam to evaluate tissue edema and skin laxity. Patients were treated with a dose of antibiotics (cephalexin 1000 mg or clindamycin 600 mg if penicillin-allergic) before the procedure. Biopsies were obtained before drug treatment initiation and within 21 days of the last dose of drug delivery. The post-treatment biopsy excluded the scar resulting from the pre-treatment biopsy site. 

Tissue biopsies were embedded in paraffin and cut into 5-µm sections for H&E and immunofluorescent staining. Sections were rehydrated, and antigen retrieval was achieved with 90 °C sodium citrate (Sigma-Aldrich, St. Louis, MO, USA). Non-specific binding was blocked with 5% donkey serum. Tissues were incubated overnight with the primary antibody at 4 °C. Primary antibodies for immunohistochemical staining included rabbit polyclonal anti-human collagen I (ab34170; Abcam, Cambridge, UK), rabbit polyclonal anti-human Ki67 (ab15580; Abcam), goat polyclonal anti-human CD4 (AF379; R&D Systems, Minneapolis, MN, USA), goat polyclonal anti-human LYVE-1 (AF2089; R&D Systems), rabbit polyclonal anti-human periostin (ab14041; Abcam), rabbit polyclonal anti-human IL13 (ab9576; Abcam), mouse monoclonal anti-human mast cell tryptase (ab2378; Abcam), rabbit polyclonal anti-human thymic stromal lymphopoietin (TSLP; ab188766; Abcam), guinea pig polyclonal anti-human cytokeratin14 (ab192694; Abcam), goat polyclonal anti-human IL33 (AF3625; R&D System), rat monoclonal anti-IL25 (NBP2-11677; Novus Biologicals, Centennial, Colorado), rabbit polyclonal anti-human IL13 receptor (ab79277; Abcam), mouse monoclonal anti-human IL4 (MAB304; R&D Systems), and mouse monoclonal anti-human IL5 (MAB605; R&D Systems). Negative controls were treated with no primary antibody.

Immunofluorescent staining was performed using Alexa Fluor fluorophore-conjugated secondary antibodies (Life Technologies, Grand Island, NY, USA). Images were scanned using the Mirax Viewer software (Carl Zeiss Microimaging; White Plains, NY, USA), and analysis was conducted using the Panoramic Viewer software (3DHISTECH, Budapest, Hungary). Cell counts and epidermal thickness analyses were performed in 4–5 randomly selected high-powered fields per patient. Collagen I, IL13, and periostin quantification was performed using the MetaMorph software (Molecular Devices, San Jose, CA, USA) to analyze staining intensity in 5 high-powered fields (20 × per specimen).

### 2.5. Statistical Analysis 

Statistical analysis was performed using the Graphpad Prism software. Volume measurements, bioimpedance, and tonometry were performed in duplicate and an average between the two measurements was used in the analyses. Quantification of histological sections was also performed using an average of 4–5 sections per sample. Normal distribution of data was assessed using the Anderson–Darling test. Normally distributed data were analyzed using paired *t*-tests. Data that were not normally distributed were analyzed using the non-parametric Wilcoxon signed rank test.

## 3. Results

### 3.1. Patient Demographics

A total of nine patients were accrued to the study during the first 3.5 months after the trial was opened (three patients in July, two patients in August, three patients in September, and one patient in October). The planned accrual was limited due to the availability of the drug and expiration deadline. Ultimately, the patient accrual was stopped due to a lack of availability of the study drug. Patient demographics are presented in Table 4. The average age of our patients was 54 ± 9 years, and these patients had lymphedema for an average duration of 7.6 ± 5 years. All patients had a history of stage II–III breast cancer, treatment with axillary lymph node dissection, and radiation therapy. Most patients were overweight (BMI 27 ± 2.7 kg/m) but not obese at the time of the trial, and the majority underwent mastectomy. Pre-enrollment lymphedema treatment for each patient is presented in Table 5. Most patients used a combination of massage/physical therapy and compression treatment. 

### 3.2. Adverse Events

There were 13 adverse events in the trial (Table 6). Most adverse events were minor and self-limited. These adverse events were evenly distributed, with each of the nine patients experiencing at least one adverse event. Two level three adverse events were noted in two individual patients. One patient with a history of cellulitis in her lymphedematous limb developed cellulitis 1 month after enrollment. This adverse event required hospitalization, and the patient was removed from the study per our protocol; the patient recovered uneventfully after intravenous antibiotic treatment. Another patient with a history of stage IIIB inflammatory breast cancer developed pulmonary metastasis during the study period. This development occurred after completing drug treatment and required removal of the patient from the washout analysis. 

### 3.3. Arm Volume and Bioimpedance Measurements

Post-treatment arm volumes were modestly, though significantly, increased compared with baseline measures (753 mL + 348 mL vs. 817 + 354 mL; *p* = 0.046; Figure 1a). The increased average volume in our cohort was primarily related to one patient who experienced a 27% increase after completion of treatment. Unfortunately, this patient developed metastatic disease approximately 3 months after the end of treatment and was removed from the study; as a result, washout arm volume measurements were not obtained for this individual. 

Bioimpedance values (L-Dex) in all patients were outside the normal range (>10) in all phases of the study (Figure 1b). There were no statistical differences between pretreatment and post-treatment, or post-treatment and washout time points. In contrast, we noted reductions in skin tonometry after treatment, and these changes persisted in the washout period (Figure 1c; *p* < 0.01 baseline vs. treatment; *p* < 0.01 baseline vs. washout).

### 3.4. Quality of Life Outcomes

Changes in lymphedema related quality of life as assessed by the ULL-27 questionnaire are presented in Figure 2. We noted improvements (increases) in physical (Figure 2a) and social domains (Figure 2b) when comparing the post-treatment scores with pretreatment values. The average increase in the physical domain was 13 points (*p* < 0.035). Similarly, the scaled scores for the social domain improved after treatment (11.9-point increase; *p* < 0.01). Improvements in both scales were lost and returned to pretreatment levels when patients were re-examined after the washout period (i.e., 4 months after the last treatment). We did not note changes in the psychological domain at any time point (Figure 2c). 

### 3.5. Histologic Analysis

We compared histologic changes in the skin of the normal and lymphedematous limbs before and after treatment with QBX258. We found no histologic changes in the skin of the normal limb following drug treatment (not shown); in contrast, histologic and immunofluorescent analysis of skin biopsies obtained from the same anatomic location of the lymphedematous limb demonstrated marked changes following drug treatment. 

#### QBX258 Treatment Decreases Hyperkeratosis and Fibrosis 

Biopsy specimens obtained from the lymphedematous limb before treatment with QBX258 demonstrated thickening of the epidermis (hyperkeratosis) compared with the normal limb (Figure 3a; *p* < 0.02). Consistent with this observation, we noted an increased number of proliferating (Ki67^+^) keratinocytes in the basal layer of the skin (Figure 3b; *p* < 0.01). We also found that the expression of cytokeratin 14 (KRT14), an intermediate filament protein that is expressed by mitotically active, less differentiated keratinocytes typically located in the basal layer of the skin, was present in nearly the entire thickness of the epidermis in the lymphedematous samples (Figure 3c; *p* < 0.05). Interestingly, we found that after treatment with QBX258, these pathological changes improved in the lymphedematous limb, and, for the most part, returned to normal levels noted in the unaffected limb (hyperkeratosis, *p* < 0.01; Ki67+ cells, *p* < 0.05; KRT14 *p* < 0.001). In contrast, we found no differences in these parameters in the normal limb skin following treatment with QBX258 (not shown).

Consistent with our previous reports, we found that the deposition of type I and type III collagen was increased in the papillary and, to a lesser extent, in the reticular dermis of lymphedema biopsy specimens before drug treatment compared with the unaffected limb (Figure 4a–d; Collagen I papillary dermis, *p* < 0.01; Collagen I reticular dermis, *p* < 0.05; Collagen III papillary dermis, *p* = NS; Collagen III reticular dermis, *p* = NS). Treatment with QBX258 decreased type I collagen deposition to near-normal levels in the papillary and reticular dermis in the lymphedema limb biopsy specimens (but did not change type I collagen expression in the skin of the unaffected limb [not shown]); however, there was some variability in this response, but these changes did not reach statistical significance (Figure 4a,b). In contrast, we noted a more consistent response in type III collagen deposition with decreased protein expression in both the papillary and reticular dermis (Figure 4c,d; collagen III papillary dermis, *p* = 0.01; collagen III reticular dermis, *p* = 0.02).

Periostin, also known as osteoblast-specific factor 2, is an extracellular matrix protein that can modulate immune responses and regulate tissue remodeling in skin wound healing and inflammation [40]. Periostin is highly expressed in inflammatory skin conditions such as atopic dermatitis and increases proliferation and differentiation of keratinocytes. Increased periostin expression is associated with skin and lung fibrosis [40,41]. Consistent with these observations, we noted that periostin expression was increased in the basal layer of the epidermis of the skin biopsies obtained from the lymphedematous limb compared with the normal limb (Figure 4e, *p* = 0.02). QBX258 treatment modestly decreased periostin expression in this histologic area in the lymphedematous but not the normal limb; however, this change did not reach statistical significance (Figure 4e and not shown). 

Consistent with our previous clinical biopsy and laboratory findings [14], we found that the number of CD4^+^ cells was increased (nearly 2-fold; *p* < 0.01) in the skin/subcutaneous tissues of the lymphedematous skin relative to the normal limb before treatment with QBX258 (Figure 5a). Infiltrating CD4^+^ cells were primarily localized to the reticular dermis and clustered around mesenchymal cells and lymphatic vessels in this area. We also found that the number of putative Th2 cells, CD4^+^ cells that expressed Th2 cytokines IL4 and IL5, was increased in the lymphedematous limb compared with normal skin (Appendix A; *p* < 0.05 for CD4^+^IL4^+^ cells; *p* < 0.001 for CD4^+^IL5^+^ cells). Treatment with QBX258 decreased the number of CD4^+^ cells in the lymphedematous skin; however, this change did not reach statistical significance. In contrast, the number of CD4^+^IL5^+^ cells was decreased and approached the levels noted in normal skin following treatment with QBX258 (Appendix A; *p* = 0.01). The number of CD4^+^IL4^+^ cells was also decreased; however, this difference did not quite reach statistical significance (*p* = 0.06).

Mast cells have also been implicated in inflammatory skin conditions and contribute to pathological changes by producing various cytokines. Mast cells are characterized by secretory granules containing proteases, including tryptase and chymase [42]. Staining of normal and lymphedema skin biopsy specimens for tryptase revealed that the number of tryptase positive cells is significantly increased in lymphedematous tissues (Figure 5b). Analysis of biopsy specimens double-stained for lymphatic vessels (LYVE-1^+^), and tryptase suggested that the number of mast cells in lymphedematous tissues is increased, and that these cells, similar to CD4^+^ cells, tend to accumulate around capillary lymphatics in the lymphedematous tissues (*p* < 0.01 vs. normal skin). Interestingly, analysis of biopsy specimens after treatment with QBX258 demonstrated a decrease in the number of infiltrating mast cells in the lymphedematous but not normal tissues (*p* < 0.05; Figure 5b and not shown). 

The expression of Th2-inducing cytokines, such as TSLP (thymic stromal lymphopoietin), IL33, and IL25, is increased in atopic dermatitis, and other Th2-mediated disorders [43,44,45,46]. These cytokines promote Th2 responses by modulating activation of antigen-presenting cells that promote Th2 biased differentiation. Consistent with this, we found that the expression of TSLP, IL33, and IL25 was increased in the superficial layer of the epidermis compared with normal tissues (Figure 6a–c). While the expression of these cytokines was largely absent in normal skin keratinocytes, we noted strong cytoplasmic expression of TSLP and IL25 and nuclear expression of IL33 in the keratinocytes located in the stratum granulosum. Quantification of the tissue area with positive staining revealed a 12.8-, 3.8-, and 6.8-fold increase in TSLP, IL33, and IL25 expression, respectively (TSLP, *p* < 0.001; IL33, *p* < 0.01; IL25, *p* < 0.001). This expression pattern was mirrored by the expression of the IL13 receptor (IL13R) with high levels of expression (6.9-fold increase; *p* < 0.001) noted in the superficial layer of the epidermis compared with normal skin. Interestingly, treatment with QBX258 significantly decreased, and in some cases normalized, the expression of Th2-inducing cytokines and IL13R by keratinocytes (TSLP, *p* < 0.001; IL33, *p* < 0.001; IL25, *p* < 0.02; IL13R, *p* < 0.01). 

## 4. Discussion

In this pilot study, we utilized an immunotherapy approach to treat secondary lymphedema and found that these treatments are well-tolerated, accrual to a study of this type is feasible, and that patients are highly motivated to participate. Given the nature of this disease and our many exclusion criteria, we were pleased with the accrual of patients to the trial and would have been successful in recruiting our goal number had we not been limited by the drug expiration. Although our patients experienced some adverse events, these issues were mostly self-limited and minor. One patient developed lymphedema-related cellulitis approximately 6 weeks after starting therapy; however, this patient had a history of cellulitis in the past (most recently 6 months before the study). Recurrent cellulitis occurs in about 30% of patients with lymphedema and is a significant cause of morbidity in this patient population. Nevertheless, the patient recovered after intravenous antibiotics. 

One patient, who had a history of locally advanced breast cancer, developed pulmonary metastasis after completing drug therapy in our trial. Although causality cannot be ruled out, given the aggressive nature of her underlying disease process this event is unfortunately not unexpected. Indeed, before the start of our trial, we were very concerned about the potential for immunotherapy to increase the risk of cancer recurrence or tumor metastasis since T cell responses play a key role in tumor immune responses [47,48]. However, previous clinical and experimental studies suggest that Th2 cytokines promote tumor growth and metastasis and that inhibition of Th2 differentiation may potentiate tumor immune responses [47,49,50]. Breast tumors are infiltrated by Th2 cells that strongly express IL4 and IL13, and these cytokines promote tumor growth by directly interacting with tumor cells, inhibiting DC responses and regulating the expression of cancer cell differentiation markers. Activation of IL4/IL13 signaling pathways increases breast cancer invasion and propensity for lung metastasis [51]. Importantly, the blockade of Th2 responses increases tumor surveillance by immune cells and inhibits tumor growth and metastasis in various tumor types, including breast cancer [47,48,52,53,54]. These findings led some authors to propose that the blockade of Th2 responses may be a viable strategy for breast cancer treatment suggesting that these treatments would also be safe for the treatment of BCRL [55]. 

A disappointing finding in our study was that QBX258 treatment was not effective in decreasing arm volumes (in fact, arm volumes were slightly increased after treatment). This result may be a statistical error because we did not reach our accrual goal. It is also possible that this increase in volume was related to avoiding manual lymphatic massage or high compression garments during the study period. It is also possible that our inclusion criteria of a relatively large baseline volume differential (>300 cc) biased our selection towards patients with more advanced lymphedema in whom drug treatment may be less effective or who may require longer treatments to promote reversal of the chronic pathologic changes. Fibroadipose deposition in these patients, such as end-stage changes in other fibrotic disorders, may not be reversible. This hypothesis is supported by the finding that surgical treatments for lymphedema are most effective for patients with early-stage (stage 0 or 1) disease [56]. Our findings suggest that future trials may be more meaningful if they are geared towards patients with an earlier-stage disease or if more sensitive primary outputs (e.g., patient-reported outcomes, skin tonometry, L-Dex, and histologic examination) are selected. 

Other investigators have also reported on the use of anti-inflammatory treatments in patients with lymphedema. For example, Rockson and colleagues used mouse models of lymphedema to show that the molecular profile of the disease is characterized by inflammation, fibrosis, and oxidative stress [57]. In subsequent preclinical studies, these authors found that treatment with non-steroidal anti-inflammatory medications (ketoprofen) decreases inflammation and improves lymphedema [13,57]. These findings led to a clinical trial to test the efficacy of ketoprofen for the treatment of lymphedema. The study design included an initial open-label treatment phase (21 patients) followed by a randomized controlled study in which 34 patients with either primary or secondary lymphedema of the upper or lower extremity were treated with ketoprofen twice daily (16 patients) or placebo (18 patients) for 4 months [58]. This study, similar to ours, showed that treatment with ketoprofen improved skin histologic examination (decreased thickness and improved histopathology score) and decreased skin infiltration with macrophages and neutrophils. However, similar to the findings of our current study, ketoprofen treatment did not reduce the overall limb volumes or decrease tissue fluid content as assessed by bioimpedance measurements. More recent studies from Rockson and colleagues have shown that the therapeutic benefits of ketoprofen are attributable to the inhibition of leukotriene B_4_ (LTB_4_) [59]. Bestatin, an LTB_4_ inhibitor, improved histologic findings of lymphedema, decreased inflammation, and enhanced lymphatic function in a mouse model of lymphedema. Interestingly, low concentrations of LTB_4_ improved lymphangiogenesis; however, higher concentrations of this molecule, as found in patients with lymphedema, inhibit lymphangiogenesis and lymphatic vessel sprouting. A recent phase 2 clinical trial with Bestatin (ULTRA trial) finished enrollment, and molecular studies of treatment responses are underway (Stanley Rockson, personal communication). 

Several recent studies have shown that doxycycline is effective for the treatment of secondary lymphedema caused by filarial infections and that these improvements may be related to the anti-Th2 effects of this drug [20,60]. For example, in a clinical trial of 162 patients randomized to treatment with either amoxicillin (control group), doxycycline, or placebo for 6 weeks, found that patients treated with doxycycline had significant reductions in the severity of lymphedema 12 and 24 months after treatment. Nearly 44% of patients treated with doxycycline had decreased lymphedema stage at these time points; in contrast improvements were only noted in 3.2% and 5.6% of the patients treated with amoxicillin or placebo, respectively [60]. In support of our findings suggesting that Th2 cytokines play a role in the pathophysiology of secondary lymphedema, a more recent report showed that improvements resulting from doxycycline treatment were related to decreased Th2 inflammatory responses in a mouse model of filariasis [20]. Taken together, these findings suggest that secondary lymphedema resulting from either surgical injury or helminth infection and lymphatic obstruction may share a common pathophysiology and that treatments aimed at this pathway may be effective. 

Patients with lymphedema have a substantially decreased quality of life [61] and, to our knowledge, ours is the first study in which patient-reported outcomes were analyzed using a validated questionnaire following drug treatment. We noted significant improvements in the physical and social but not psychological dimensions of our quality of life measures after treatment with QBX258. These improvements were statistically significant overall, with five out of eight patients reporting improved outcomes. Improvements with drug therapy disappeared, returning to baseline levels in the washout period after the drug was discontinued. Although it is possible that improvements in some patients were simply a placebo effect due to the study design, it is also possible that these results reflect subtle improvements in pathological changes that are not reflected by gross changes such as limb volume measurements. Another possibility is that changes in the psychological dimension are less responsive to treatment due to the chronic nature of the disease process. Our findings are consistent with Toyserkani et al., who recently reported their results on a preliminary study on the use of adipose-derived regenerative cells to treat BCRL in 10 patients [62]. In this study, women with unilateral stage I or II BCRL with an average excess volume of 300 cc were treated with adipose-derived stem cells injected into the axillary area and evaluated with measurements and quality of life questionnaires (disabilities of the arm, shoulder, and hand [DASH] outcome and Lymphedema Quality of Life [LYMQOL]) 1, 3, and 6 months after treatment. Similar to the findings in our study, Toyserkani et al. found that stem cell treatments did not decrease arm volume or dual energy x-ray absorptiometry (DXA); however, the authors noted significant improvements in physical findings, such as a sensation of heaviness or tension, as well as the overall DASH scores. Interestingly, changes in the psychological aspects of lymphedema, as assessed by LYMQOL, did not improve with stem cell therapy, suggesting that these impairments may be more challenging to treat or may reflect complex psychological issues related to cancer diagnosis and treatment. 

Our hypothesis that treatment with QBX258 results in subtle but significant tissue changes that may be responsible for improvements in quality of life scales and decreased skin fibrosis as reflected by skin tonometry is supported by our histologic analysis. In these studies, we noted significant decreases in hyperkeratosis, epidermal proliferation, type III collagen deposition, and a decrease in the number of infiltrating mast cells in biopsy samples obtained from the lymphedematous arm compared with the normal contralateral limb. Indeed, consistent with our findings with skin tonometry, we found that the thickness of the epidermis in the lymphedematous limb decreased to levels that were equivalent to the normal arm after 4 months of treatment with QBX258. We also noted decreases in the number of CD4^+^ cells and type I collagen deposition. However, these differences did not reach statistical significance, possibly because our patient accrual was lower than expected. These findings are supported by Rockson et al., who showed that treatment with ketoprofen improved pathological skin changes in lymphedema [58]. Our findings are also supported by our previous studies in which we have shown surgical treatment of lymphedema not only improves the symptoms of lymphedema but also improves pathological skin changes [63].

Interestingly, we found that patients with lymphedema have significantly increased expression of Th2-inducing cytokines (TSLP, IL33, IL25) in the epidermal cells of the lymphedematous limb, and that treatment with QBX258 decreased or normalized these changes. This finding is important and suggests that epidermal cells may play a role in the pathophysiology of lymphedema and may help coordinate or initiate chronic Th2 responses. This hypothesis is supported by previous studies implicating epidermal cell expression of these cytokines in the pathophysiology of other atopic diseases, including atopic dermatitis, asthma, allergic rhinoconjunctivitis, and eosinophilic esophagitis [43,44,45,46,64,65]. 

TSLP is a member of the IL2 cytokine family and is expressed primarily by epithelial cells in the skin, lungs, and the gut. TSLP expression can be induced by bacterial products, inflammatory cytokines, or injury and is required for antiparasitic responses by regulating Th2 cytokine expression [66,67]. In addition, TSLP has been implicated in breast cancer pathogenesis by regulating the DC expression of OX40L, differentiating the Th2 cells, and increasing production of IL13 [47]. Consistent with our findings in lymphedematous skin, keratinocytes are significant producers of TSLP, and the expression of TSLP increased the skin lesions of patients with atopic dermatitis [68]. Experimental overexpression of TSLP in keratinocytes or intradermal injections of this cytokine, by acting on DCs and other mechanisms, result in the development of atopic dermatitis in mice, infiltration of Th2 differentiated CD4^+^ cells, and systemic Th2 inflammatory responses [67,68,69,70,71]. TSLP also acts synergistically with IL1 and TNF to activate mast cells that produce high levels of Th2 cytokines [66]. Mast cells, in turn, can regulate epithelial TSLP expression in allergic rhinitis [44]. The role of mast cells in lymphedema pathophysiology remains unknown; however, our findings that mast cell numbers are increased in lymphedematous skin and return to normal levels following QBX258 treatment suggest that these cells may play a role in this disease. 

Several clinical trials have targeted TSLP. For example, Tezepelumab is a monoclonal human IgG2 that neutralizes TSLP by interfering with its receptor binding. A phase 2 clinical trial showed that treatment with Tezepelumab decreased the rates of asthma exacerbations in patients with moderate to severe disease over a 1-year period [72]. This drug also showed improvement in a study of 113 adults with atopic dermatitis over a 12-week period, but this result did not achieve statistical significance due to greater than expected improvements in placebo-treated patients [73]. Our study’s findings suggest that TSLP may also be a target for lymphedema treatment and that additional studies are warranted. 

IL33 is a member of the IL1 family of cytokines and, similar to TSLP, has been implicated in various atopic diseases including atopic dermatitis, eosinophilic esophagitis, and asthma by genome-wide association studies as well as by studies demonstrating increased expression of IL33 in affected tissues [45,46,64,74,75,76]. Levels of IL33 expression correlate with the severity of atopic dermatitis [77]. IL33 is expressed by epithelial cells of the skin, gut, and lung and acts on naïve CD4^+^ cells to promote Th2 differentiation and activation of mast cells and eosinophils [76,78]. Consistent with the findings from our study, IL33 is expressed in the nucleus of keratinocytes but is released in pathological conditions [78,79]. IL33 is also expressed by activated mast cells [79], and these cells can modulate IL33 activity by processing the full-length protein [80]. 

IL25, similar to IL33 and TSLP, induces Th2-biased responses, promotes the expansion of splenic plasma cells and eosinophils, and increases expression of IL4 and IL13 [81]. IL25 is thought to amplify atopic inflammatory reactions by increasing the expression of Th2 cytokines such as IL13 and IL4 [81] and can cause mast cell degranulation [82]. Our findings suggest that the expression of TSLP, IL33, and IL25 is increased in the epidermis of the lymphedematous skin and that treatment with QBX258 decreases this response. These findings require additional study and suggest that epidermal cells may play a role in the pathology of lymphedema by coordinating or amplifying Th2 responses. 

## 5. Conclusions

This study, given its modest size and short duration, has many limitations. The most important issue, particularly with regards to patient-reported outcomes, is the possibility that our results are reflective of a placebo effect. This is certainly conceivable and requires consideration in future trial designs. In addition, limitations in accuracy in measuring limb volumes together with the daily variation that many patients experience due to diet, changes in exercise, or other life events, suggests that these measurements may be less useful in future study designs. Thus, if we perform a study on patients with early-stage lymphedema in whom limb volume differences are 100–200 cc, it may be difficult to show statistically significant changes in the outcomes. Therefore, given that a significant issue with lymphedema is qualify of life [83], an expanded panel of studies in patient-reported outcomes may be more effective and informative. This approach may also be highly beneficial in prevention studies since the development of quality of life changes is more sensitive and occurs before the development of overt physical changes [84]. In conclusion, our study suggests that immunotherapy for lymphedema is well-tolerated and that patients are motivated to participate. In addition, our pilot study suggests that patient-reported outcomes and histologic changes may be more sensitive than physical measurements. 

## Figures and Tables

**Figure 1 biology-10-00934-f001:**
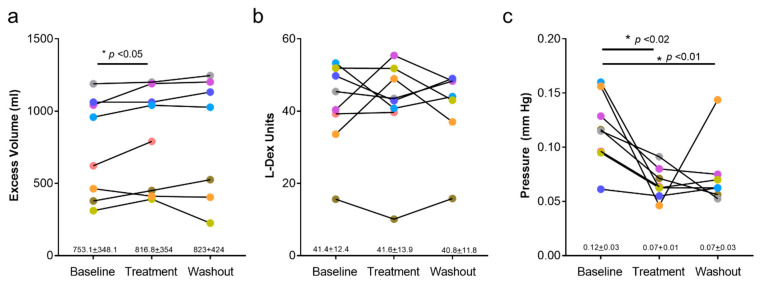
Effects of QBX258 treatment on arm volume, bioimpedance (L-Dex), and tonometry measurements. (**a**) Quantification of arm volumes at baseline, following 4-month treatment with QBX258 (treatment), and 4 months after cessation of treatment (washout). Changes in each patient are shown over time. (**b**) L-Dex measurements at baseline, following treatment, and following washout period with QBX258. (**c**) Tonometry measurements at baseline, following treatment, and following washout period with QBX258.

**Figure 2 biology-10-00934-f002:**
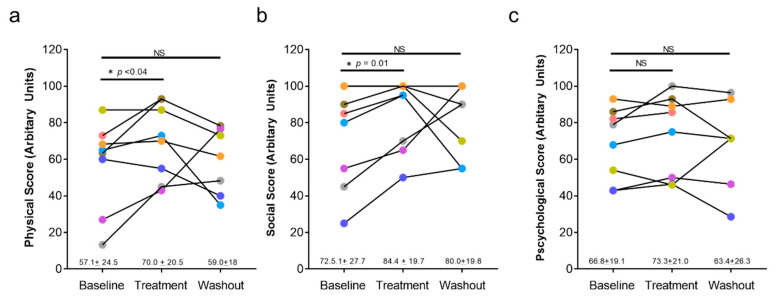
Effects of QBX258 treatment on quality of life measured using the ULL-27. (**a**) Quantification of physical score at baseline, following treatment, and following washout period in each patient over time. Average score for the cohort ± SD is shown below the graph. (**b**) Quantification of social score at each time point for all patients. (**c**) Quantification of psychological score at each time point for all patients.

**Figure 3 biology-10-00934-f003:**
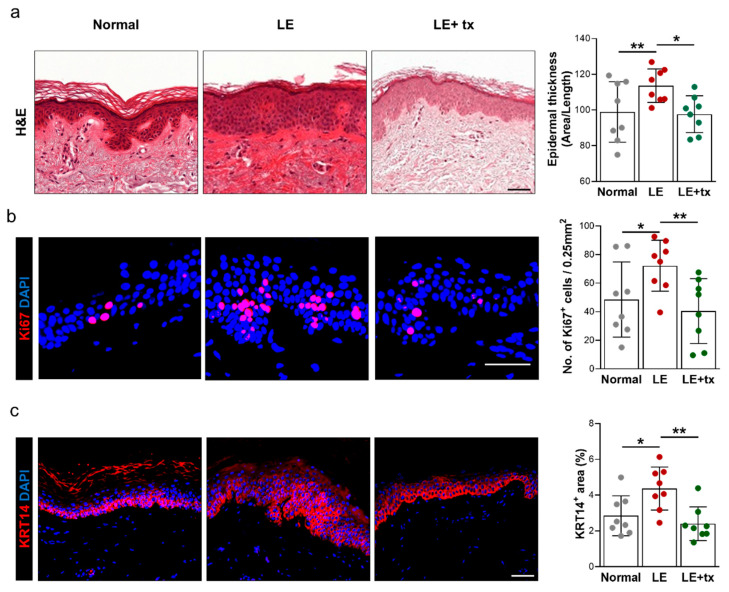
Treatment with QBX258 decreases hyperkeratosis and keratinocyte proliferation in lymphedematous skin. (**a**) Representative H&E images (right panels) and quantification (right panel; average ± SD) of matched normal, lymphedematous skin before treatment (LE), and lymphedematous skin biopsy specimens obtained 4 months after treatment with QBX258 (LE + tx). (**b**) Representative immunofluorescent staining of normal, LE, LE + tx biopsy specimens stained for DAPI (nuclear stain, blue) and Ki67 (nuclear stain, pink). (**c**) Representative immunofluorescent staining of normal, LE, LE + tx biopsy specimens stained for DAPI (nuclear stain, blue) and KRT14 (membrane stain, pink). Scale bars: 50 μm; * *p* < 0.05; ** *p* < 0.01.

**Figure 4 biology-10-00934-f004:**
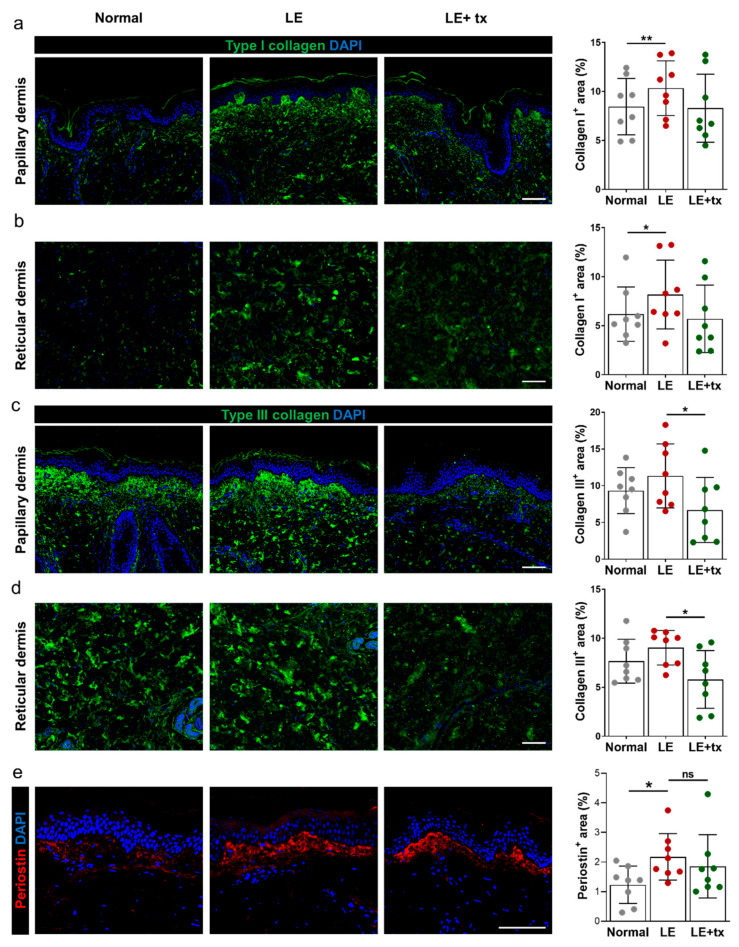
Treatment with QBX258 decreases type III collagen deposition in lymphedematous tissues. Fibrosis after QBX258 treatment. (**a**,**b**) Representative immunofluorescent staining (left panels) and quantification (right panel) in normal, LE, LE + tx biopsy specimens for DAPI (blue) and type I collagen (green) in the papillary (**a**) and reticular (**b**) dermis. (**c**,**d**) Representative immunofluorescent staining (left panels) and quantification (right panel) in normal, LE, LE + tx biopsy specimens for DAPI (blue) and type III collagen (green) in the papillary (**c**) and reticular (**d**) dermis. (**e**) Representative immunofluorescent staining (left panels) and quantification (right panel) of normal, LE, LE + tx biopsy specimens for DAPI (blue) and periostin (red). Scale bar: 50 μm; * *p* < 0.05; ** *p* < 0.01.

**Figure 5 biology-10-00934-f005:**
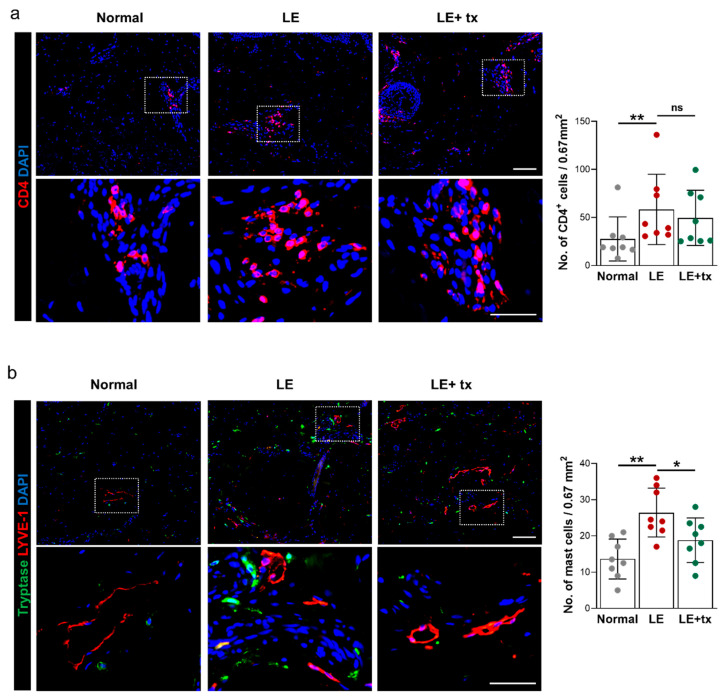
Treatment with QBX258 decreases mast cell infiltration in lymphedematous skin. (**a**) Representative low-power (upper images) and high-power (lower images) immunofluorescent staining (left panels) and quantification (right panel) of DAPI (blue) and CD4^+^ cells (pink) in normal, LE, and LE + tx biopsy specimens. Area in dotted box is shown in high-power views. (**b**) Representative low-power (upper images) and high-power (lower images) immunofluorescent staining (left panels) and quantification (right panel) of DAPI (blue), LYVE-1 (red), and tryptase (green) in normal, LE, and LE + tx biopsy specimens. Area in dotted box is shown in high-power views. Scale bar (low-power images): 200 μm; scale bar (magnified images): 25 μm; * *p* < 0.05; ** *p* < 0.01.

**Figure 6 biology-10-00934-f006:**
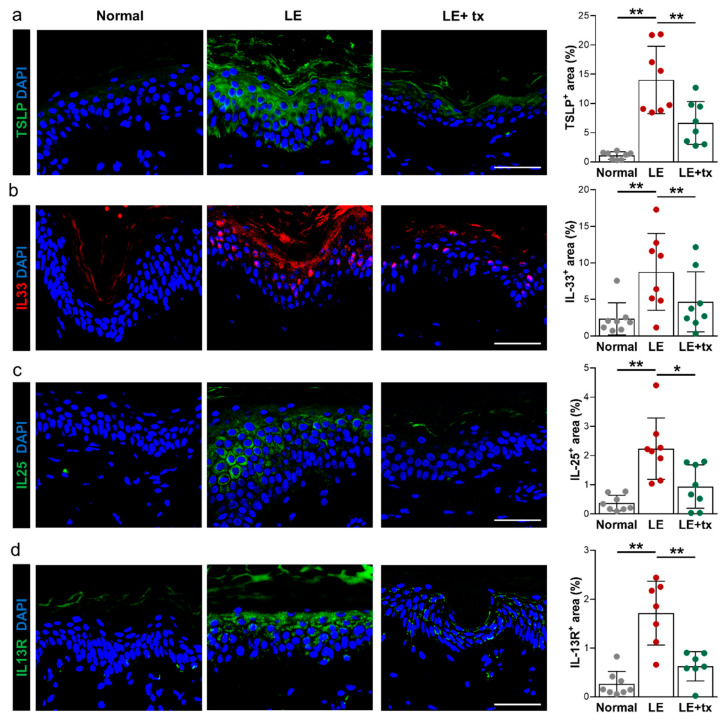
Lymphedema results in increased epidermal expression of Th2-inducing cytokines. This phenotype is mitigated by treatment with QBX258. (**a**) Representative immunofluorescent staining (left panels) and quantification (right panel) of TSLP (green) in normal, LE, and LE + tx biopsy specimens. (**b**) Representative immunofluorescent staining (left panels) and quantification (right panel) of IL33 (red) in normal, LE, and LE + tx biopsy specimens. (**c**) Representative immunofluorescent staining (left panels) and quantification (right panel) of IL25 (green) in normal, LE, and LE + tx biopsy specimens. (**d**) Representative immunofluorescent staining (left panels) and quantification (right panel) of IL13R (green) in normal, LE, and LE + tx biopsy specimens. Scale bar: 100 μm; * *p* < 0.05; ** *p* < 0.01.

**Table 1 biology-10-00934-t001:** Inclusion and exclusion criteria.

Inclusion	Exclusion
Women 18–70 with unilateral stage I or II BCRL	Bilateral lymphedema or history of bilateral axillary lymph node dissection
Volume difference of at least 300 mL between the normal and lymphedema limb	Recent (within last 3 months) history of cellulitis
BMI 18–30	Current (within last month) use of chemotherapy or radiation
No current evidence of breast cancer	Recent (within last month) or current intensive manual lymphatic massage and/or short stretch bandage use
At least 6 months postop from axillary lymph node dissection	Unstable lymphedema (i.e., worsening symptoms/measurements in the past 3 months)
	Pregnant or nursing (lactating) women
	Stage III lymphedema
	Chronic use of acetaminophen (>1 gm/day for ≥3/7 days, or >2 gm/day for ≥1 day)
	Use of other investigational drugs ≤30 days or 5 half-lives of enrollment (whichever is longer)
	History of hypersensitivity to study drugs or to drugs of similar chemical classes (e.g., monoclonal antibodies, polyclonal gamma globulin, polysorbates).

**Table 2 biology-10-00934-t002:** Study design.

Assessment	Enrollment	Infusion 1	Infusion 2	Infusion 3	Infusion 4	Outcome	Washout
	Week −2–0	Week 0	Week 4	Week 8	Week 12	Week 12–15	Week 2835
Demographics	x						
Physical exam	x	x	x	x	x	x	x
Pregnancy test	x	x	x	x	x		
ECG	x					x	
Blood tests, Urinalysis	x					x	
Arm volumes	x					x	x
Bioimpedance	x					x	x
Tonometry	x					x	x
QOL questionnaire	x					x	x
Skin biopsy	x					x	
Adverse event check	x	x	x	x	x	x	x

**Table 3 biology-10-00934-t003:** ULL-27 questionnaire.

Physical Functioning	Psychological Dimension	Social Dimension
Difficulties grasping high objects	Feeling sad	Difficulty taking advantage of good weather, such as outside the house
Difficulties maintaining certain positions	Feeling Discouraged	Difficulty with personal projects, holidays, hobbies
Arm feels heavy	Feeling a lack of self-confidence	Difficulties in emotional life with spouse or partner
Arm feels swollen	Feeling well in one’s self	Difficulty in social life
Difficulties dressing	Feeling a wish to be angry	Fearful of looking in the mirror
Difficulties going to sleep	Having confidence in the future	
Difficulties holding objects		
Difficulties walking/heavy arm		
Difficulties washing		
Difficulties taking public transport		
Tingling, burning feelings		
Feeling of swollen, hard, tense skin		
Difficulties in working relationship and tasks		

**Table 4 biology-10-00934-t004:** Patient demographics.

Patient	Age	Stage	Years with Disease	Time to Develop Lymphedema	History of Cellulitis?	BMI
1	62	II	8	~3 years	No	22.1
2	59	II	3	~8 months	No	27.5
3	59	II	5	<1 year	Yes	25.3
5	36	II	3	<1 year	No	29.7
6	49	II	9	<1 year	No	29.8
7	59	II	18	<1 year	Yes	24.7
8	46	II	11	<1 year	Yes	27.7
9	60	II	4	<1 year	Yes	29.1

**Table 5 biology-10-00934-t005:** Pre-enrollment treatment for lymphedema.

Patient	Baseline Lymphedema Treatment
1	Massage, exercise
2	Physical therapy 2 × a week, lymphedema pump (~3 × a week)
3	Sleeve at night, self-manual lymphatic massage (MLD)
5	MLD 1 × a week, ACE bandage at night, occasionally pump
6	Sleeve/glove 24 h/day, compression pump, intermittent self-MLD
7	Wrapping at nighttime, lymph node transfer
8	Compression sleeve
9	Sleeve during the day, short stretch bandage 4 nights/week, lymph node transplant

**Table 6 biology-10-00934-t006:** Adverse effects.

Adverse Event	Patients (n)
Restless legs	2
Tiredness	1
Dyspnea on exertion	1
Heartburn	1
Food poisoning	1
Rash	1
Trouble sleeping	1
Light-headedness	1
Sore throat	1
Upper lip numbness	1
Cellulitis	1
Pulmonary metastases	1

## Data Availability

Not applicable.

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
