# Peer review of "Pilot Study of Anti-Th2 Immunotherapy for the Treatment of Breast Cancer-Related Upper Extremity Lymphedema"

_biology, 2021, doi:10.3390/biology10090934_

Round 1

Reviewer 1 Report

A well-written manuscript that presents some interesting data. A few points to be considered.

Page 4, line 148. Although the next paragraph describes participant selection with respect to LE stage, A volume difference of 300 mL is larger than the more typical volume difference used as a guide to implementing LE therapy, around 150 to 200 mL. Perhaps the authors could provide justification for 300 mL.

Page 6, line 178. The bioimpedance technique is quite sensitive to standardisation of the assessment. Please comment on this, at the very least state that measurements were performed according to manufacturer's instructions. Also the impedance technique has received some adverse comment regarding sensitivity in the literature. While Ref 31 is correct perhaps the more recent review of the technique could also be cited which discusses this (DOI: 10.1089/lrb.2020.0085)

Line 184. The authors used the Delfin ElastiMeter. Had they considered also using the company's skin moisture meter? TDC measurements are proving very useful in local LE assessment.

Line 230. Was limb dominance taken into account when assessing inter-limb volume differences? The L-Dex system (impedance measurements) does.

Table 2. Since patients returned to the clinic for infusions why were measurements such as perometry, tonometry and impedance not performed at each time-point? They are quick and simple to perform and the consecutive data to assess trend would have been a more powerful design.

In addition, do the authors have reliability data for test-retest over time? For example, differences in volume or tonicity from enrollment to outcome could be simply due to measurement differences. Related to this, there are no control patients who did not receive treatment. Is the wash-out period meant to represent internal control? How do the authors ensure that any changes observed are not simply a reflection of normal disease progression and/or fluctuation?

Line 296. Since measurements were obtained at 3 time points, ANOVAR with post hoc testing would be more appropriate than simple paired t-tests.

Line 301. A key point. The sample size is very small, 9 patients. Was any power calculation [performed to assess confidence in ability to assess significant change)? The sample is also quite heterogeneous (Tables 4 & 5).

Fig 1. It would be useful to identify patients by different symbols for each patient. This would allow,  for example, to illustrate whether or not there was congruence between volume changes and changes in L-Dex score since both are assumed to be measuring the same thing, changes in lymph (extracellular water) volume. If they disagree then this requires discussion. Similarly for Fig 2 to use the same symbols.

Histology. I am always slightly nervous about reliance on "representative" images. Could all images be made available as supplementary data on-line for interested readers? Also please make it clear that the histograms are for all patients (I assume) not values from the images shown.

Discussion

Line 535. Anti-inflammatory treatment may have wide ranging metabolic effects. In metabolic syndrome,  widely regarded as involving inflammatory processes, there are changes in adipose tissue deposition. Is it possible that the failure to detect volume changes could be due to the effect of treatment upon the progression of "watery" LE to the more fibrotic and adipose-laden LE, i.e. the volume doesn't change but the compositional makeup of the volume changes?

The Discussion reads well but may be considered overlong. In places, it reads like a review of the field rather than being targeted to placing the study findings in context.

Lines 529 and 624. The authors pick up my point above regarding the 300 mL volume difference. - good.

Author Response

Thank you for your careful analysis. Please find the attached file. 

Reviewer 2 Report

This is a well designed and well documented pilot study of anti-Th2 immunotherapy for the treatment of breast cancer–related lymphedema. Congratulations for your results, and hope to see larger population study.

Author Response

We thank you for your careful analysis. Please do not hesitate to contact me if any additional information is required.

Reviewer 3 Report

Authors suggested that QBX258 (combination of IL4/13 neutralizing antibody) could be potential immunotherapy for lymphedema. However, author also mentioned the loopholes of the study in discussion and conclusion sections. The manuscript is well written. Mostly easy to follow and the figures are well put together. This study gives the hope to improve the quality of life for lymphedema containing patients.

The quality of figures and table presented by authors is satisfactory.

The statistical methods are applied correctly.

Supplementary material is missing.

What is authors opinion about testing cytokines (IL4/IL13) levels in patients (before and after treatment with QBX258) serum samples?

Author Response

Q) Supplementary material is missing.

A) We will make sure supplementary material is uploaded.

Q)What is authors opinion about testing cytokines (IL4/IL13) levels in patients (before and after treatment with QBX258) serum samples?

A) This is a good idea; however, there are diurnal variations in serum cytokine levels and we have previously found that we need a large number of patients to obtain statistical significance and chose not to do this here (cuzzone,et al., 2016 Please find the attachment)
